# Calcium-Based Imaging of the Spine at Dual-Layer CT and Evaluation of Vertebral Fractures in Multiple Myeloma

**DOI:** 10.3390/cancers16152688

**Published:** 2024-07-28

**Authors:** Simone C. Brandelik, Stefanie Rahn, Maximilian Merz, Wolfram Stiller, Stephan Skornitzke, Claudius Melzig, Hans-Ulrich Kauczor, Tim F. Weber, Thuy D. Do

**Affiliations:** 1Clinic of Diagnostic and Interventional Radiology (DIR), Heidelberg University Hospital, 69120 Heidelberg, Germany; simone.brandelik@med.uni-heidelberg.de (S.C.B.); stefanie.rahn@med.uni-heidelberg.de (S.R.); wolfram.stiller@med.uni-heidelberg.de (W.S.); stephan.skornitzke@gmail.com (S.S.); claudius.melzig@med.uni-heidelberg.de (C.M.); hans-ulrich.kauczor@med.uni-heidelberg.de (H.-U.K.); tim.weber@med.uni-heidelberg.de (T.F.W.); 2Department for Hematology, Cell Therapy and Hemostaseology, University Hospital Leipzig, 04103 Leipzig, Germany; maximilian.merz@medizin.uni-leipzig.de

**Keywords:** computed tomography, multiple myeloma, spine, fractures, dual-layer spectral CT

## Abstract

**Simple Summary:**

The purpose of the study is to evaluate the prediction of vertebral fractures in patients with plasma cell dyscrasias using dual-layer CT. The study retrospectively enrolled 81 patients diagnosed with plasma cell dyscrasia who underwent whole-body DLCT at diagnosis and follow-up. In the baseline CT, conventional CT image data (CI), calcium-suppressed image data (CaSupp25 and CaSupp100), and virtual calcium-only (VCa) image data were quantitatively analyzed in the lumbar vertebral bodies and in fractured vertebral bodies. New vertebral fractures occurred in 24 patients during follow-up. Lower CT numbers in CI and VCa, and higher CT numbers in CaSupp25, at baseline were significantly associated with a higher risk of new fractures. Logistic regression analysis revealed that CT numbers in CaSupp25 and VCa might be better predictors of fractures than CI. Cut-off values were suggested: CI at 103 HU and VCa at 129 HU. In conclusion, quantitative assessment with CaSupp and calculation of VCa can predict vertebral fracture risk in multiple myeloma patients, indicating that DLCT may be useful in detecting imminent fractures.

**Abstract:**

Purpose: To evaluate the prediction of vertebral fractures in plasma cell dyscrasias using dual-layer CT (DLCT) with quantitative assessment of conventional CT image data (CI), calcium suppressed image data (CaSupp), and calculation of virtual calcium-only (VCa) image data. Material and Methods: Patients (*n* = 81) with the diagnosis of a plasma cell dyscrasia and whole-body DLCT at the time of diagnosis and follow-up were retrospectively enrolled. CI, CaSupp25, and CaSupp100 were quantitatively analyzed using regions of interest in the lumbar vertebral bodies and fractured vertebral bodies on baseline or follow-up imaging. VCa were calculated by subtraction (CaSupp100-CaSupp25), delineating bone only. Logistic regression analyses were performed to assess the possibility of imminent spine fractures. Results: In 24 patients, new vertebral fractures were observed in the follow-up imaging. The possibility of new vertebral fractures was significant for baseline assessment of CT numbers in CI, CaSupp25, and VCa (*p* = 0.01, respectively), with a higher risk for new fractures in the case of lower CT numbers in CI (Odds ratio = [0.969; 0.994]) and VCa (Odds ratio = [0.978; 0.995]) and in the case of higher CT numbers in CaSupp 25 (Odds ratio 1.015 [1.006; 1.026]). Direct model comparisons implied that CT numbers in CaSupp 25 and VCa might show better fracture prediction than those in CI (R^2^ = 0.18 both vs. 0.15; AICc = 91.95, 91.79 vs. 93.62), suggesting cut-off values for CI at 103 HU (sensitivity: 54.2%; specificity: 82.5; AUC: 0.69), for VCa at 129 HU (sensitivity: 41.7%; specificity: 94.7; AUC: 0.72). Conclusions: Quantitative assessment with CaSupp and calculation of VCa is feasible to predict the vertebral fracture risk in MM patients. DLCT may prove useful in detecting imminent fractures.

## 1. Introduction

The plasma cell dyscrasia multiple myeloma (MM) is characterized by a monoclonal proliferation of plasma cells that predominantly takes place in the bone marrow of the spine as the primary site of hematopoiesis. Changes in the microenvironment of the affected bone marrow lead to disease progression, and finally to focal osteolysis and general reduction of bone density [1,2]. Subsequent pathological vertebral fractures producing substantial pain, and at worst-case spinal cord injury, are frequent complications and contribute to morbidity and mortality in MM [3]. Whole-body low-dose CT is routinely performed in plasma cell dyscrasias to detect osteolysis, giving an overview of the condition of the skeleton simultaneously [4]. Bisphosphonates and Denosumab are listed among substances for MM patients to attenuate MM-related bone disease [3]. Other stabilizing treatment options are support by a corset, radiotherapy, cement augmentation, and spondylodesis [3,5]. MM patients with surgical intervention have a high risk of perioperative complications [3]. Identifying imminent vertebral body fractures, as the most frequent site for fractures in multiple myeloma, would be preferable and could contribute to targeted conservative precautions such as radiotherapy or support by a corset to reduce morbidity.

The introduction of a novel CT system utilizing a dual-layer detector technique allows for retrospective evaluation of spectral data, leveraging two detector layers that absorb X-ray photons at different energy levels. The top layer, composed of an yttrium-based garnet scintillator, selectively absorbs low-energy photons, while high-energy photons pass through to the bottom layer, made of gadolinium oxysulphide, where they are absorbed and converted into light. Unlike dual-energy dual-source CT, this system requires only one X-ray tube and does not necessitate the selection of an acquisition mode. Tube potentials for clinical use are set at either 120 kVp or 100 kVp, and spectral data are consistently available as both detector layers contribute to image reconstruction. Spectral analysis and postprocessing algorithms can be applied to all CT data, provided that spectral raw data sets are archived. From these spectral data, virtual non-calcium (VNCa) images can be generated to subtract calcium-based attenuation, facilitating the evaluation of bone marrow. Thus, in dual-layer spectral computed tomography (DLCT), the detector-based measurement of different parts of the polychromatic patient-attenuated X-ray spectrum allows retrospective spectral analyses and the selective depiction or suppression of materials. Clinical investigations utilizing non-contrast CT and contrast-enhanced dual-layer CT with virtual monochromatic imaging have demonstrated the potential for bone mineral density (BMD) estimation and opportunistic osteoporosis screening. However, these methods may still be susceptible to inaccuracies due to the influence of bone marrow effects, as previously noted [6,7].

In virtual non-calcium (VNCa) images, the osseous component is removed to a certain degree depending on the selected calcium suppression (CaSupp) index. The VNCa technique in DLCT and dual-source dual-energy CT has demonstrated improvements in separation of normal and metastatic bone in solid malignancies [8], as well as feasibility to differentiate between non-infiltrated and infiltrated bone marrow in MM [9,10] and also to assess tumor burden in MM [11]. Spectral data in DLCT have also shown improved accuracy for diagnosing acute osteoporotic vertebral fractures compared to conventional CT [12]. Bone mineral density (BMD) assessments on conventional CT images may yield lower values due to the interference from surrounding soft tissue components [13,14]. Calculated virtual calcium-only (VCa) CT depicting only the osseous components has been useful in detecting osteoporosis, which was shown in a previous study [15]. VCa images demonstrated superior concordance with DXA measurements and established calcium concentrations compared to CI, suggesting their utility in estimating bone mineral density (BMD). A VCa threshold of 126 HU was identified as a potential marker for detecting abnormal BMD.

The aim of this study is to evaluate calcium-based imaging of the spine for the possibility of fracture risk of vertebral bodies in patients with MM using VNCa image data and calculated virtual calcium-only (VCa) CT image data in routinely performed whole-body DLCT.

## 2. Material and Methods

### 2.1. Ethics Approval and Consent

This retrospective exploratory single-center study was approved by the local institutional review board (application number: S-348/2019). The need for written informed consent was waived.

### 2.2. Clinical Data Selection and Study Design

For study inclusion, patients were screened in the hospital and radiological information system (I.S.-H.*med., SAP; Centricity RIS-i, GE Healthcare, Chicago, IL, USA). The diagnosis of plasma cell dyscrasia was established based on the criteria of the International Myeloma Working Group including evidence of end-organ damage or amyloidosis, the percentage of plasma cell infiltration in bone marrow biopsy, serum and urinary monoclonal protein, the serum involved/uninvolved free light chain ratio, and baseline imaging [16]. Patients fulfilling the following criteria were included in the study: Patients with routinely performed non-contrast whole-body low-dose DLCT and non-contrast whole-body magnetic resonance tomography at initial diagnosis of a plasma cell dyscrasia (multiple myeloma, smoldering multiple myeloma, or monoclonal gammopathy of unknown significance) from August 2018 to August 2021 (baseline CT) and any available follow-up imaging of the spine (CT, MRI, or X-ray) until March 2022. Exclusion criteria were prior anti-myeloma chemotherapeutic treatment, other malignant diseases, missing spectral data, and long-segment spondylodesis of the lumbar spine in baseline CT. The inclusion criteria were matched by 81 patients. Follow-up imaging was CT in 60, MRI in 18, and X-ray in 2 patients. In one patient, for follow-up the spine was assessed partly by CT and partly by MRI. Follow-up imaging involved the whole spine in 74 cases and was restricted to the thoracic spine in 3, to the lumbar spine in 2, and to thoracic/lumbar spine in 2 patients.

### 2.3. DLCT Acquisition, Post-Processing and Image Analysis

Non-contrast CT acquisitions were performed in helical mode from the vertex of the skull to the knees using a dual-layer detector technique (IQon Spectral CT, Philips, Amsterdam, The Netherlands). Acquisition parameters were the same as in a previous study [17]: Tube potential 120 kV_p_, dose right index 15 (automated attenuation-based dose modulation), average tube current-time product 93 mAs, volumetric computed tomography dose index (CTDI_vol_) 8.4 mGy, pitch 1.0, gantry rotation time of 0.75 s and collimation 64 × 0.625 mm. Dual-layer data of all examinations were stored routinely. After retrospective inclusion into the study, VNCa image reconstructions of baseline DLCT data and CT number measurements were performed with the manufacturer’s dedicated image post-processing software (IntelliSpace Portal Version 11, Philips). CT numbers, also known as CT attenuation values, are quantified using the Hounsfield scale, where the radiodensity of distilled water at standard pressure and temperature (STP) is set to 0 Hounsfield Units (HUs). VNCa images were reconstructed using CaSupp indices 25 and 100 in sagittal orientation. Reconstruction slice thickness was 1 mm. For measuring the respective CT numbers (mean value and SD), oval regions of interest (ROIs) were positioned manually in the vertebral bodies L1–L5 in all patients and additionally in all vertebral bodies of the cervical and thoracic spine that presented with fractures on the baseline or the patient’s follow-up imaging. L1–L5 were chosen for measuring in all patients because these are often affected by fractures and are the largest vertebrae of the spine allowing most reliable measures. ROI sizes were chosen as large as reasonably possible with a 2 mm distance to the cortex and the dorsal vertebral vein area, avoiding large focal lesions and sclerosis if possible. Vertebrae without residual spongiosa due to cement augmentation, spondylodesis, or high-grade fracture were excluded. ROIs were copied to all reconstructions, ensuring intraindividual comparability. Two radiologists performed the ROI placement independently (6 and 10 years of experience).

Virtual calcium-only image data were obtained by subtraction of the CaSupp 25 from the CaSupp 100 (Figure 1).

The grading of the fracture degree of the vertebral bodies was also assessed in both baseline and follow-up: The most severe point of cortical fracture (the point of most narrowing of the upper and lower corticalis) served for classification into grades 1–3 (<25%, 25–75%, >75% of the original height). Secondary sintering of preexisting fractures between baseline and follow-up was noted down.

### 2.4. Statistical Analysis

For statistical analysis, R version 4.2.0 was used [18]. The data distribution was evaluated by descriptive statistics and graphics (quantile–quantile plots, histograms, and kurtosis). Model assumptions were checked prior to the main statistical analyses according to standard procedures. Statistical significance was set at *p* < 0.05. To assess the possibility of vertebral fractures in follow-up imaging based on the baseline assessment, logistic regression models were used with grand mean centering of continuous and effect coding of dichotomous predictors [19]. The odds ratio (OR) was calculated for each predictor. For the major analyses, a mean attenuation value for the whole lumbar spine was computed for each participant and used as the main predictor in regression analyses. The odds ratios (ORs) derived from logistic regression analyses quantify the change in odds associated with a one-unit variation in the predictor variable, specifically baseline CT numbers. Given that CT numbers were expressed in Hounsfield Units (HUs) across all reconstruction methods, the reported ORs correspond to a per-1 HU increment.

The Bonferroni–Holm method was used for the correction of multiple comparisons [20]. Akaike’s second-order corrected information criterion and R² were applied for the comparison of different regression models [21].

For the evaluation of inter-reader reliability, the intraclass correlation coefficient (ICC) was calculated for the CT numbers of each lumbar vertebra.

The diagnostic performance of CI and VCa to identify patients with fractures was assessed using receiver operating characteristic (ROC) curve analysis. The area under the ROC curve (AUC) was quantified [22]. Finally, a cut-off value for CI and VCa with the best overall diagnostic performance was selected using the Youden index [23].

## 3. Results

### 3.1. Descriptive Statistics

The mean age of all patients (*n* = 81, 31 female, 50 male) at the time of baseline DLCT was 61.15 years. The mean time between baseline and follow-up imaging was 15.1 months (range 1–32 months).

Vertebral body fractures at baseline were observed in 43 patients (53.1%), the mean number of fractures per patient was 4.6, mean fracture degree of the whole spine was 1.6. Twenty-four patients (29.6%) showed new vertebral fractures at follow-up, the mean number of new fractures was 2.3, mean fracture degree was 1.8. Secondary sintering of preexisting fractures between baseline and follow-up was seen in 23 patients (28.4%). For descriptive results, see also (Table 1).

Most vertebral fractures were found in the lower thoracic and lumbar spine: maximum in L3 (*n* = 18), followed by Th8, Th 11, and L1 (each *n* = 16 patients).

Inter-rater agreement was high for mean CT attenuation values in conventional CT (ICC 0.97 [0.95, 0.98]) and in VCa (ICC 0.96 [0.93, 0.97]) and also for the respective measurements of single lumbar vertebrae (Table 2).

### 3.2. Probability of New Vertebral Fractures

The occurrence of new vertebral fractures at follow-up was significantly associated with baseline CT measurements of the lumbar spine in conventional CT, CaSupp 25 VNCa images, and VCa calculations (*p* = 0.01 for all). Specifically, patients presenting with lower CT numbers in conventional CT (OR: 0.982 [0.969; 0.994]) and in VCa calculations (OR: 0.987 [0.978; 0.995]) were at a significantly higher risk for developing new fractures. Conversely, patients with higher CT numbers in CaSupp 25 images (OR: 1.015 [1.006; 1.026]) also exhibited a significantly increased risk for new vertebral fractures. These findings underscore the prognostic value of baseline CT metrics in predicting fracture risk and highlight the need for careful interpretation of these imaging parameters in clinical practice. The incidence of a vertebral fracture was significantly predicted by CT numbers of the lumbar spine in conventional CT, CaSupp 25 VNCa images, and in VCa (*p* = 0.01, respectively). The comparison of model fit parameters revealed that models with CaSupp 25 and VCa CT numbers showed consistently higher R² and lower AICc compared to the model with conventional CT numbers (R^2^ = 0.18 both vs. 0.15; AICc = 91.95 and 91.79 vs. 93.62), which might imply a better model fit for VCa and Ca Supp 25.

Further exploration, using CT attenuation from single lumbar vertebrae as the predictor, showed that fracture predictability for any new fracture in the whole spine was best for CT numbers in L3 (R^2^ = 0.18; AICc = 91.72), followed by L4 (R^2^ = 0.17; AICc = 92.45), and L2 and L5 (R^2^ = 0.16 each; AICc = 93.08 and 93.09, respectively).

### 3.3. Exploratory Analysis of Further Predictive Parameters

Multiple logistic regression analysis for identifying the criteria for the possibility of new vertebral fractures in VCa calculation revealed statistical significance just for the CT number at baseline (*p* = 0.02) but neither for age, sex, the time between baseline and follow-up nor the presence of a fracture at baseline (Table 3). After backward elimination of predictors, the predictors “CT number at baseline” (*p* = 0.01; odds ratio 0.989 [0.979;0.997]) and “presence of a fracture at baseline” (*p* = 0.07; odds ratio 2.795 [0.940;9.085]) remained. Therefore, the presence of a former vertebral fracture might also be of predictive relevance for the incidence of new fractures.

Multiple logistic regression analysis and backward elimination for conventional CT and CaSupp 25 images also revealed statistical significance for the possibility of new vertebral fractures just for the predictor “CT number at baseline”, but with slightly lower R^2^ and higher AICc than for VCa calculation (Table 4).

ROC curve analysis with Youden’s index suggests an optimal cut-off value for CI at 103 HU (sensitivity: 54.2%; specificity: 82.5%) with an AUC of 0.69 (Figure 2A) and a cut-off value for VCa at 129 HU (sensitivity: 41.7%; specificity: 94.7%) with an AUC of 0.72 (Figure 2B).

## 4. Discussion

To the best of our knowledge, this study is the first to assess the possibility of vertebral fractures in patients with plasma cell diseases on baseline DLCT, analyzing conventional, virtual non-calcium, and virtual calcium-only data. Spectral data of DLCT provide the opportunity to analyze selected parts of the material composition of the bone and to give insight into the myeloma-related bone disease. Dependent on the selected CaSupp index, VNCa imaging allows variably profound analyses of the bone marrow [16]. Several studies have already performed such VNCa-based analyses of bone marrow edema in fractures [12,24,25]. However, this study investigates the calcium parts of the bone, which are the target of osteolytic destruction. A novel VNCa imaging-based calculation of hypothetical virtual calcium-only (VCa) data is analyzed for the prediction of imminent fractures on baseline DLCT in MM patients. Our results show a significantly higher risk for new fractures in patients with lower CT numbers in conventional CT and in VCa calculation—representing a lower bone density, and in patients with higher CT numbers in CaSupp 25, representing a higher degree of bone marrow infiltration—and meet our expectations based on the material composition in conventional, VNCa, and VCa data. These results illustrate the bone structure-weakening interaction of plasma cells with the bone marrow microenvironment on imaging [1,2]. Interestingly, the possibility of new fractures was comparable for spectral CT data and conventional CT, however, slightly better for spectral CT data. This demonstrates that focusing on the calcium level and the degree of bone marrow infiltration may be beneficial in future screening for imminent pathological fractures of the spine.

Quantitative assessment of VNCa images in DLCT for the discrimination between diffuse and non-diffuse infiltration in MM has already been shown to be feasible [17]. In conventional CT, visual assessment of diffuse bone marrow infiltration can be difficult up to impossible. Still, spectral CT data revealed to enable to determine the extent of bone marrow infiltration [17].

As myeloma-related bone disease often manifests with (multi)focal osteolysis and not only with general osteopenia, in MM whole-body DLCT will predict the fracture risk superior to the routine dual-energy X-ray absorptiometry (DEXA) known from the osteoporosis work-up that examines femur and lumbar spine. Additionally, DEXA performance is limited by local density increase due to spine interventions, fractures, or degeneration [26], alterations that can better be evaluated on cross-sectional imaging. But DEXA may be replaced in the future by DLCT. A phantom study found that bone mineral density can be quantified with high accuracy on various DLCT protocols [27], and another study depicted the feasibility to obtain bone marrow density information from non-dedicated contrast-enhanced DLCT examinations [7], both showing the further potential of DLCT in this field of interest.

As a limitation, due to the retrospective study design, the mean time between baseline and follow-up imaging was 15.1 months with a relatively wide range (1–32 months). In a prospective design, patients could have undergone their follow-up after a specific time interval or at relapse of first-line therapy. Additionally, several previously determined follow-ups could have been performed and analyzed. However, the favored imaging technique for these investigative follow-ups would be MRI to avoid unnecessary radiation exposure, raising challenges such as costs, claustrophobia, implants, and length of examination in anguished patients. Two of our patients received X-ray as follow-up. Due to higher sensitivity, in a prospective study, follow-up should be performed using cross-sectional imaging.

A further study analyzing the fracture possibility of a “second” baseline DLCT at the time of the first relapse with an additional follow-up would be of interest. At this later time of the disease course, myeloma-related bone disease would be more advanced, possibly providing new aspects for DLCT evaluation, and the possibility of imminent fractures would be even more critical.

A noted limitation of this methodology is its reduced efficacy in identifying osteoblastic processes. This includes rare mixed subtypes of multiple myeloma with osteosclerotic lesions [28,29], particularly in cases associated with POEMS syndrome [30] osteoblastic metastases, or metabolic diseases such as Paget’s disease. Given our specific aim to investigate myelomatous mineral bone, and to enhance the comparability and interpretability of CT measurements, regions of interest (ROIs) were meticulously selected to avoid large focal lesions and sclerosis whenever feasible. This methodological approach was intended to standardize the assessment; however, it may inadvertently result in the underestimation or overestimation of fracture risk in vertebrae exhibiting substantial focal lesions or sclerosis. This limitation is significant as it could impact the accuracy of fracture risk predictions and underscores the need for cautious interpretation of CT data in these cases.

Additionally, the study has limitations due to the lack of correlation between CT-detected myeloma lesions and histological analysis or DEXA scans, which are the gold standard for assessing bone mineral density. This gap restricts the direct validation of CT findings against the most reliable benchmarks for bone mineral assessment. However, all patients in the study underwent MRI examinations at the time of initial diagnosis. MRI, known for its superior soft-tissue contrast and ability to detect marrow infiltration, provides additional diagnostic information that may help mitigate this limitation. This approach has been included to enhance the study’s robustness and address potential gaps in data correlation.

The principal limitation of this study is the absence of a fracture probability assessment based on new fractures in multiple myeloma vertebrae observed between baseline and follow-up in 24 patients, as the low sample size precluded a statistically robust analysis.

In addition, MM research has revealed a wide inter- and intraindividual heterogeneity of the plasma cell genetics and the composition of the surrounding bone marrow microenvironment [31]. Such differences were also detected between osteolytic lesions and infiltrated but non-osteolytic bone areas [32] and between newly diagnosed and relapsed/refractory MM [33]. Consequently, these factors contribute to disease progression, including the extent of destruction of mineral bone, and also influence possibilities of treatment and administered medication. The prediction of fractures by DLCT is certainly influenced by the individual speediness of disease progression and treatment options.

For future studies, an even newer technique may also be considered for investigation: In a study analyzing conventional CT images in MM patients, dual-source photon-counting CT turned out to be superior to dual-source dual-energy CT regarding spatial resolution of bony microstructure and lytic lesions [34]. Dual-source and spectral photon-counting CT are a promising radiation dose-efficient technique for future patient care in MM [35,36] and the prediction of vertebral fractures in MM.

In summary, this study presents a quantitative approach with a high inter-rater agreement for the evaluation of fracture risk of vertebral bodies in DLCT examinations of patients with MM.

## 5. Conclusions

Quantitative assessment of spectral DLCT data with VNCa image data and calculation of virtual calcium-only data show promising results for fracture risk evaluation of vertebral bodies in MM patients. The possibility of new vertebral fractures between baseline and follow-up DLCT was significant for conventional CT, CaSupp 25 VNCa image data, and VCa calculation with spectral data tending to higher possibility. DLCT may prove to be useful in detecting imminent fractures and prevent fracture-related morbidity.

## Figures and Tables

**Figure 1 cancers-16-02688-f001:**
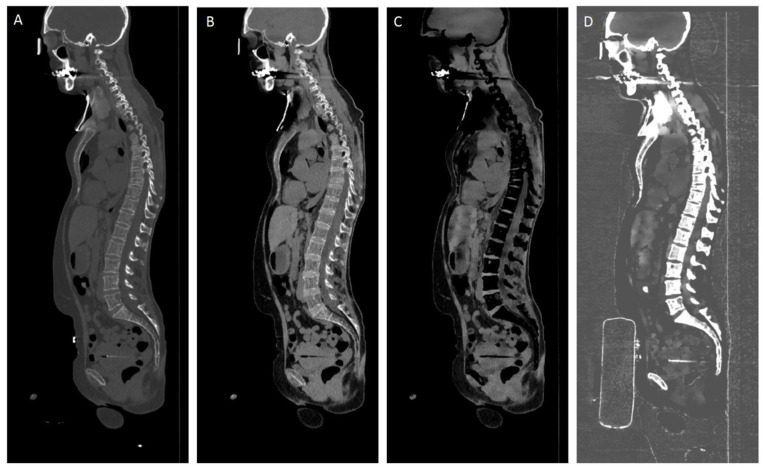
Sagittal reformations of whole-body CT in a multiple myeloma patient. Conventional CT image (**A**), virtual non-calcium (VNCa) image using CaSupp index 25 (**B**) and CaSupp index 100 (**C**), and virtual calcium-only (VCa) image obtained by subtraction of the CaSupp 25 from the CaSupp 100 (**D**).

**Figure 2 cancers-16-02688-f002:**
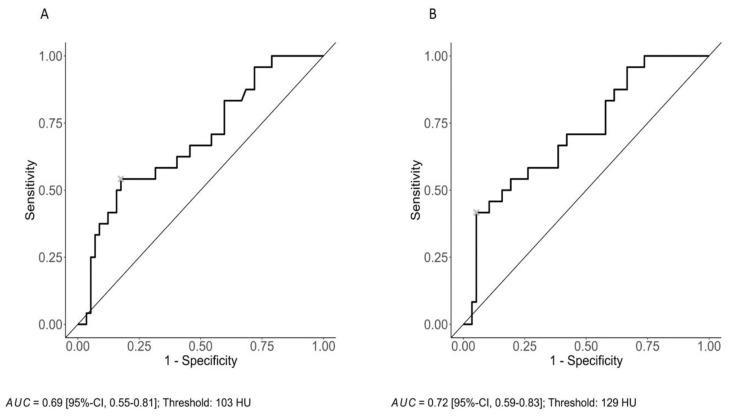
ROC curve analysis with an optimal cut-off value for CI at 103 HU (sensitivity: 54.2%; specificity: 82.5%) with an AUC of 0.69 (**A**) and a cut-off value for VCa at 129 HU (sensitivity: 41.7%; specificity: 94.7%) with an AUC of 0.72 (**B**).

**Table 1 cancers-16-02688-t001:** Patient characteristics.

	Total Sample (*n* = 81)
age (years)	
*M* (*SD*)	61.1 (10.9)
sex (*n* (%))	
female	31 (38.27%)
male	50 (61.73%)
time between baseline and follow-up assessment (months)	
*M* (*SD*)	15.08 (8.23)
patients with surgical intervention of the spine (*n* (%))	
baseline	8 (9.88%)
follow-up	14 (17.28%)
fractures of the spine at baseline assessment	
number of patients with fractures (*n* (%))	43 (53.09%)
average number of fractures (*M* (*SD*))	4.63 (3.86)
average fracture degree	1.62 (0.53)
new fractures of the spine at follow-up assessment	
number of patients with new fractures (*n* (%))	24 (29.63%)
average number of new fractures (*M* (*SD*))	2.33 (1.71)
average fracture degree	1.80 (0.53)
sintering of fractures between baseline and follow-up assessment	
number of patients with sintered fractures (*n* (%))	23 (28.40%)
average number of sintered fractures (*M* (*SD*))	2.00 (1.24)
average fracture gradient after sintering (*M* (*SD*))	2.08 (0.57)

**Table 2 cancers-16-02688-t002:** Intra-Class Correlations (ICC) for measurements of CT attenuation values of the lumbar spine for conventional image data (CI) and virtual calcium-only (VCa) image data basing on single-rater [k = 2], absolute agreement, 2-way random-effects models.

	ICC [95% CI]	
	CI	VCa
Vertebral Body		
L1	0.92 [0.88, 0.95]	0.86 [0.80, 0.91]
L2	0.94 [0.91, 0.96]	0.94 [0.91, 0.96]
L3	0.91 [0.86, 0.94]	0.79 [0.69, 0.86]
L4	0.80 [0.70, 0.87]	0.93 [0.90, 0.96]
L5	0.93 [0.89, 0.95]	0.93 [0.90, 0.96]
*N* = 81		

**Table 3 cancers-16-02688-t003:** Results of multiple logistic regression analysis for the possibility of spinal fracture at follow-up (FU) from CT attenuation of the lumbar spine in virtual calcium-only images (VCa) at baseline (BL).

	b (SE)	*p*	Odds Ratio	95% CI for Odds Ratio
Intercept	−1.06 (0.30)	<0.01		
CT attenuation of LS at baseline in VCa	−0.01 (0.00)	0.02	0.989	[0.979; 0.997]
age	0.01 (0.03)	0.70	1.010	[0.960; 1.063]
sex	0.50 (0.55)	0.37	1.642	[0.552; 4.905]
time between BL and FU	−0.01 (0.04)	0.80	0.991	[0.922; 1.065]
fracture at BL	0.98 (0.63)	0.12	2.661	[0.801; 9.758]
R² = 0.25, AICc = 96.27, Model χ^2^(5) = 15.31, *p* < 0.01				

**Table 4 cancers-16-02688-t004:** Model fit parameters Akaike’s second-order corrected information criterion and R² for conventional image data (CI), virtual calcium-only (VCa), and image data with calcium suppression index 25 (CaSupp25).

	R²	AICc
Forced Entry		
CI	0.23	97.20
VCa	0.25	96.27
CaSupp25	0.24	96.52

## Data Availability

The data that support the findings of this study are available on request from the corresponding author (T.D.D.). The data are not publicly available due to restrictions, e.g., their containing information that could compromise the privacy of patients.

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
