# Peer review of "Calcium-Based Imaging of the Spine at Dual-Layer CT and Evaluation of Vertebral Fractures in Multiple Myeloma"

_cancers, 2024, doi:10.3390/cancers16152688_

Round 1

Reviewer 1 Report

Comments and Suggestions for Authors

This is a retrospective analysis into the potential role of using dual layer CT to predict fracture risks in myeloma. Whilst interesting, I found it somewhat confusing and difficult to follow, in part due to the absence of definitions of some of the terminology used. For example- CT numbers- what do these actually refer to, how are they calculated? It would also be useful for authors to describe the difference between dual-layer CT and dual energy CT for the readers who are unfamiliar with the technique, and what are currently utilized in clinical practice. The readers are informed by the analysis that the possibility of new vertebral fractures was significant for baseline CT numbers of the lumbar spine- can this be elaborated to explain what this means?

It would also be useful to describe what other factors may affect the interpretation - do other conditions such as Paget's disease influence results?

Overall, an interesting paper but clarification would certainly be helpful.

Comments on the Quality of English Language

Minor editing required to improve clarify please. 

Author Response

We would like to thank the reviewers for providing us with these detailed and helpful comments and we would like to thank the editor for allowing us to submit a revised version of our manuscript. We have revised the manuscript based on the reviewer’s comments and we hope to have done so in a satisfactory manner. To facilitate an easy review of the changes, we have made all changes to the manuscript with tracked-changes and responded to each of the reviewer’s comments in point-by-point fashion below.

Reviewer 1:

This is a retrospective analysis into the potential role of using dual layer CT to predict fracture risks in myeloma. Whilst interesting, I found it somewhat confusing and difficult to follow, in part due to the absence of definitions of some of the terminology used. For example- CT numbers- what do these actually refer to, how are they calculated?

In the revised manuscript, we have enormously elaborated on specific terminology and revised sections to enhance clarity and precision, especially in the introduction section und discussion section.

It would also be useful for authors to describe the difference between dual-layer CT and dual energy CT for the readers who are unfamiliar with the technique, and what are currently utilized in clinical practice.

We have provided a more detailed explanation of the Dual-layer CT as follows:

„The introduction of a novel CT system utilizing a dual-layer detector technique allows for retrospective evaluation of spectral data, leveraging two detector layers that absorb X-ray photons at different energy levels. The top layer, composed of an yttrium-based garnet scintillator, selectively absorbs low-energy photons, while high-energy photons pass through to the bottom layer, made of gadolinium oxysulphide, where they are absorbed and converted into light. Unlike dual-energy dual-source CT, this system requires only one X-ray tube and does not necessitate the selection of an acquisition mode. Tube potentials for clinical use are set at either 120 kVp or 100 kVp, and spectral data is consistently available as both detector layers contribute to image reconstruction. Spectral analysis and postprocessing algorithms can be applied to all CT data, provided that spectral raw data sets are archived. From this spectral data, virtual non-calcium (VNCa) images can be generated to subtract calcium-based attenuation, facilitating the evaluation of bone marrow.”

The readers are informed by the analysis that the possibility of new vertebral fractures was significant for baseline CT numbers of the lumbar spine- can this be elaborated to explain what this means?

Thank you for the comment. We have revised this part and hope that it is now expressed more clearly. 3.2. was changed as follows:

“Probability of New Vertebral Fractures

The occurrence of new vertebral fractures at follow-up was significantly associated with baseline CT measurements of the lumbar spine in conventional CT, CaSupp 25 VNCa images, and VCa calculations (p=0.01 for all). Specifically, patients presenting with lower CT numbers in conventional CT (OR: 0.982 [0.969; 0.994]) and in VCa calculations (OR: 0.987 [0.978; 0.995]) were at a significantly higher risk for developing new fractures. Conversely, patients with higher CT numbers in CaSupp 25 images (OR: 1.015 [1.006; 1.026]) also exhibited a significantly increased risk for new vertebral fractures. These findings underscore the prognostic value of baseline CT metrics in predicting fracture risk and highlight the need for careful interpretation of these imaging parameters in clinical practice.The occurrence of new vertebral fractures at follow-up was significantly associated with baseline CT measurements of the lumbar spine in conventional CT, CaSupp 25 VNCa images, and VCa calculations (p=0.01 for all). Specifically, patients presenting with lower CT numbers in conventional CT (OR: 0.982 [0.969; 0.994]) and in VCa calculations (OR: 0.987 [0.978; 0.995]) were at a significantly higher risk for developing new fractures. Conversely, patients with higher CT numbers in CaSupp 25 images (OR: 1.015 [1.006; 1.026]) also exhibited a significantly increased risk for new vertebral fractures. These findings underscore the prognostic value of baseline CT metrics in predicting fracture risk and highlight the need for careful interpretation of these imaging parameters in clinical practice.”

It would also be useful to describe what other factors may affect the interpretation - do other conditions such as Paget's disease influence results?

Thank you for your valuable comment. We agree with the reviewer and have added this as limitation of the methodology.

“A noted limitation of this methodology is its reduced efficacy in identifying osteoblastic processes. This includes rare mixed subtypes of multiple myeloma with osteosclerotic lesions, particularly in cases associated with POEMS syndrome, osteoblastic metastases, or metabolic diseases such as Paget's disease.“

Overall, an interesting paper but clarification would certainly be helpful.

Reviewer 2 Report

Comments and Suggestions for Authors

The authors investigated calcium-based imaging of the spine by dual-layer CT retrospectively. See comments below.

1. The biggest criticism is that we cannot know whether these methods are used for myeloma bone lesions. The authors evaluated the fracture grading score by various imaging, but we can not know whether MM bone lesions are there. Though these methods seemed to be well applied to osteoporosis, we want myeloma specific data or methods. Myeloma has osteolytic lesions with low calcium contents. Can you evaluate Ca-based imaging ?

 2. Line 186188 Are these odds ratio per 1 HU ? Please show or specify.

3. Figure2 Title and legend are missing.

Author Response

We would like to thank the reviewers for providing us with these detailed and helpful comments and we would like to thank the editor for allowing us to submit a revised version of our manuscript. We have revised the manuscript based on the reviewer’s comments and we hope to have done so in a satisfactory manner. To facilitate an easy review of the changes, we have made all changes to the manuscript with tracked-changes and responded to each of the reviewer’s comments in point-by-point fashion below.

Reviewer 2

The authors investigated calcium-based imaging of the spine by dual-layer CT retrospectively. See comments below.

  1. The biggest criticism is that we cannot know whether these methods are used for myeloma bone lesions. The authors evaluated the fracture grading score by various imaging, but we can not know whether MM bone lesions are there. Though these methods seemed to be well applied to osteoporosis, we want myeloma specific data or methods. Myeloma has osteolytic lesions with low calcium contents. Can you evaluate Ca-based imaging ?

Thank you for your valuable comment. We acknowledge that a limitation of the study is the lack of correlation between CT-identified myeloma lesions and histology or DEXA. However, at our study center, all multiple myeloma patients also undergo a synchronous MRI at the time of initial diagnosis and CT. This information has been added to the Materials and Methods section.

Additionally, we have included this as a limitation of our study and have discussed it as follows:

„Given our specific aim to investigate myelomatous mineral bone, and to enhance the comparability and interpretability of CT measurements, regions of interest (ROIs) were meticulously selected to avoid large focal lesions and sclerosis whenever feasible. This methodological approach was intended to standardize the assessment; however, it may inadvertently result in the underestimation or overestimation of fracture risk in vertebrae exhibiting substantial focal lesions or sclerosis. This limitation is significant as it could impact the accuracy of fracture risk predictions and underscores the need for cautious interpretation of CT data in these cases.

Furthermore, the study is limited by the absence of a correlation between CT-identified myeloma lesions and histological analysis or DEXA scans, the latter being the gold standard for evaluating bone mineral density. The lack of these correlations represents a significant limitation, as it restricts the ability to directly validate the CT findings against the most reliable benchmarks for bone mineral assessment. Nevertheless, all patients in this study underwent concurrent MRI examinations at the time of initial diagnosis. This simultaneous imaging approach with MRI, known for its superior soft-tissue contrast and ability to detect marrow infiltration, may help to mitigate the aforementioned limitation by providing additional diagnostic information and improving the overall assessment of myeloma-related bone pathology. This methodological detail has been incorporated to enhance the study's robustness and address potential gaps in data correlation.“

  1. Line 186-188 Are these odds ratio per 1 HU ? Please show or specify.

The odds ratios (ORs) calculated within logistic regression analyses indicate the change in odds when the predictor (baseline CT numbers) changes by one unit.  As CT numbers were calculated in HU – for all reconstructions –, the given ORs are per 1 HU.

Within our study, the ORs can be understood as the factor by which the odds of a new vertebral fracture in-/decrease as CT numbers of the spine increase by 1 HU. The odds are defined as:

Odds =  p(new fracture at baseline) / p(no new fracture at baseline)

*p = probability

For example, we found an OR of 0,987 for fracture prediction with CT numbers measured in VCa reconstructions. This means that as CT attenuation in VCa increases by 1 HU, the odds of a new vertebral fracture decrease by factor 0,987.

Therefore, we have revised the explanation in the section statistical analysis as follows:

“The odds ratios (ORs) derived from logistic regression analyses quantify the change in odds associated with a one-unit variation in the predictor variable, specifically baseline CT numbers. Given that CT numbers were expressed in Hounsfield Units (HU) across all reconstruction methods, the reported ORs correspond to a per 1 HU increment.“

And we have added this phrase into the result section as follows:

“The incidence of a vertebral fracture was significantly predicted by CT numbers of the lumbar spine in conventional CT, CaSupp 25 VNCa images and in VCa (p=0.01, respectively).”

  1. Figure2 Title and legend are missing.

We apologize for the error. Thank you for bringing it to our attention. We have now added the title and legend.

Editor

Pease also address the below comments:
Add the Simple Summary, Funding, DAS part during revision as stated in the 
manuscript.

We have included a concise summary (under 200 words), funding information, and DAS statements in the revised manuscript.

Extend the Ref to 30+ and please note that the self-citation rate of the 
author's article needs to be controlled below 15%. Extend the main text to 3000+ words.

With the additional explanation und elaborate descpription in the introduction and revision of the discussion section we have achieved 35 references.

The self-citation rate is at 6%.

The main text has been extended to 3330 words.

Round 2

Reviewer 1 Report

Comments and Suggestions for Authors

Much improved. My comments have been addressed. Thank you

Author Response

Thank you for the review. No changes or answers are necessary for Round 2.

Reviewer 2 Report

Comments and Suggestions for Authors

The authors revised the manuscript. The authors do not answer the biggest criticism clearly. The authors said, “all multiple myeloma patients also undergo a synchronous MRI at the time of initial diagnosis and CT.” Therefore, the authors can discriminate between MM bone lesions and the other and analyze them separately. The authors should whether the fracture was on the MM bone lesion or other at least.

However, the authors said, “ROI sizes were chosen as large as reasonably possible with a 2 mm distance to the cortex and the dorsal vertebral vein area, avoiding large focal lesions and sclerosis if possible.” Therefore, this study has limitations whether you did not analyze the MM bone lesions though they are the most vulnerable lesions. Your study has these limitations. The amended discussion is roundabout. Please describe at discussion session more directly and clearly. 

Author Response

We would like to thank the reviewer for re-evaluating the manuscript and, above all, for his in-depth consideration of the project. We have revised and modified the manuscript accordingly.

The authors revised the manuscript. The authors do not answer the biggest criticism clearly. The authors said, “all multiple myeloma patients also undergo a synchronous MRI at the time of initial diagnosis and CT.” Therefore, the authors can discriminate between MM bone lesions and the other and analyze them separately. The authors should whether the fracture was on the MM bone lesion or other at least. 
However, the authors said, “ROI sizes were chosen as large as reasonably possible with a 2 mm distance to the cortex and the dorsal vertebral vein area, avoiding large focal lesions and sclerosis if possible.” Therefore, this study has limitations whether you did not analyze the MM bone lesions though they are the most vulnerable lesions. Your study has these limitations. The amended discussion is roundabout.

Thank you for the insightful comment. We fully concur with the reviewer's observation regarding this study limitation. While we scanned the entire study population for new fractures occurring in vertebrae with multiple myeloma lesions, the low incidence of new fractures (n=24) precluded a statistically robust analysis, as advised by our statistical experts. Consequently, we have identified this as the most significant limitation of our study and have duly noted it as follows:

„The principal limitation of this study is the absence of a fracture probability assessment based on new fractures in multiple myeloma vertebrae observed between baseline and follow-up in 24 patients, as the low sample size precluded a statistically robust analysis “.

Please describe at discussion session more directly and clearly. 

Thank you for your comment. We have revised the discussion in accordance with the recommendations provided by both the editor and Reviewer 1.

But to improve clarity we have paraphrased the most important section of the discussion as follows:

„Additionally, the study has limitations due to the lack of correlation between CT-detected myeloma lesions and histological analysis or DEXA scans, which are the gold standard for assessing bone mineral density. This gap restricts the direct validation of CT findings against the most reliable benchmarks for bone mineral assessment. However, all patients in the study underwent MRI examinations at the time of initial diagnosis. MRI, known for its superior soft-tissue contrast and ability to detect marrow infiltration, provides additional diagnostic information that may help mitigate this limitation. This approach has been included to enhance the study's robustness and address potential gaps in data correlation.

The principal limitation of this study is the absence of a fracture probability assessment based on new fractures in multiple myeloma vertebrae observed between baseline and follow-up in 24 patients, as the low sample size precluded a statistically robust analysis.“